# Invertible Consistency Distillation for Text-Guided Image Editing in Around 7 Steps

**Nikita Starodubcev**[1,2]     **Mikhail Khoroshikh**[1,2]     **Artem Babenko**[1]     **Dmitry Baranchuk**[1]

[1]Yandex Research          [2]HSE University

https://yandex-research.github.io/invertible-cd

Original                    Editing (~ **0.9 secs**)

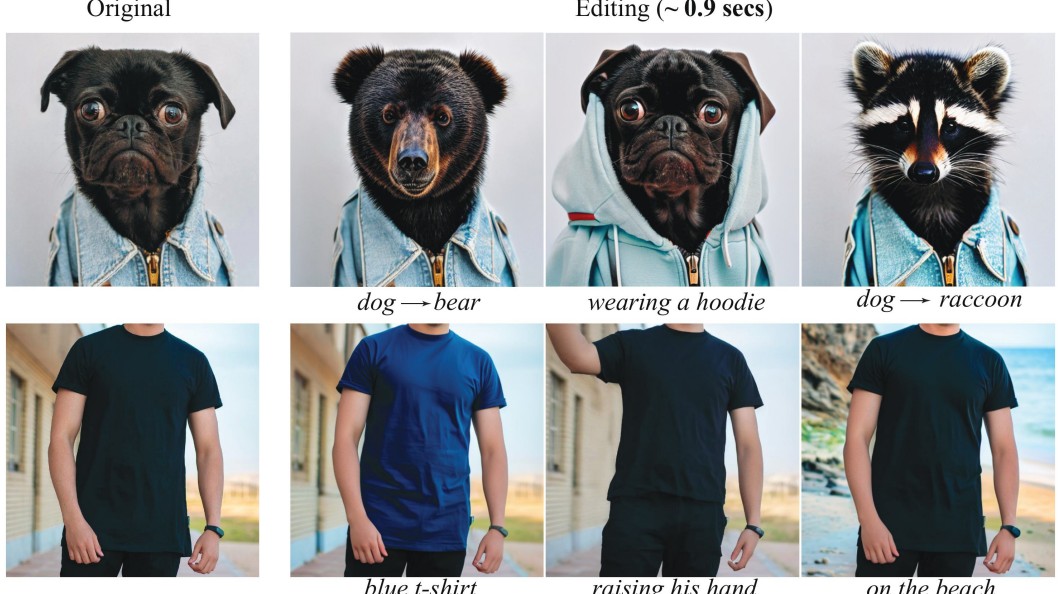

$dog \rightarrow bear$        *wearing a hoodie*        $dog \rightarrow raccoon$

*blue t-shirt*        *raising his hand*        *on the beach*

Figure 1: Invertible Consistency Distillation (iCD) enables both fast image editing and strong generation performance in a few model evaluations.

## Abstract

Diffusion distillation represents a highly promising direction for achieving faithful text-to-image generation in a few sampling steps. However, despite recent successes, existing distilled models still do not provide the full spectrum of diffusion abilities, such as real image inversion, which enables many precise image manipulation methods. This work aims to enrich distilled text-to-image diffusion models with the ability to effectively encode real images into their latent space. To this end, we introduce *invertible Consistency Distillation* (iCD), a generalized consistency distillation framework that facilitates both high-quality image synthesis and accurate image encoding in only $3-4$ inference steps. Though the inversion problem for text-to-image diffusion models gets exacerbated by high classifier-free guidance scales, we notice that *dynamic guidance* significantly reduces reconstruction errors without noticeable degradation in generation performance. As a result, we demonstrate that iCD equipped with dynamic guidance may serve as a highly effective tool for zero-shot text-guided image editing, competing with more expensive state-of-the-art alternatives.

38th Conference on Neural Information Processing Systems (NeurIPS 2024).

# 1 Introduction

Recently, text-to-image diffusion models [1, 2, 3, 4, 5, 6] have become a dominant paradigm in image generation based on user-provided textual prompts. The exceptional quality of these models makes them a valuable tool for graphics editors, especially for various image manipulation tasks [7, 8, 9]. In practice, however, the applicability of diffusion models is often hindered by their slow inference, which stems from a sequential sampling procedure, gradually recovering images from pure noise.

To speed-up the inference, many recent works aim to reduce the number of diffusion steps via diffusion distillation [10, 11, 12, 13, 14, 15, 16, 17] that has provided significant progress in high-quality generation in $1-4$ steps and has already been successfully scaled to the state-of-the-art text-to-image diffusion models [18, 19, 20, 21, 22, 23, 24]. Though the existing distillation approaches still often trade either mode coverage or image quality for few-step inference, the proposed models can already be feasible for practical applications, such as text-driven image editing [25, 26, 27].

The most effective diffusion-based editing methods typically require encoding real images into the latent space of a diffusion model. For "undistilled" models, this encoding is possible by virtue of the connection of diffusion modeling [28] with denoising score matching [29] through SDE and probability flow ODE (PF ODE) [30]. The ODE perspective of diffusion models reveals their *reversibility*, i.e., the ability to encode a real image into the model latent space and closely reconstruct it with minimal changes. This ability is successfully exploited in various applications, such as text-driven image editing [31, 32, 33], domain translation [34, 9], style transfer [35].

Nevertheless, it remains unclear if distilled models can be enriched with such reversibility since existing diffusion distillation methods primarily focus on achieving efficient generation. This work positively answers this question by proposing *invertible Consistency Distillation* (iCD), a generalized consistency modeling framework [10, 12, 13] enabling both high-quality image generation and accurate inversion in a few sampling steps.

In practice, text-to-image models leverage classifier-free guidance (CFG) [36], which is crucial for high-fidelity text-to-image generation [1, 3] and text-guided editing [32, 33]. However, the guided diffusion processes yield significant challenges for inversion-based editing methods [32]. Previous approaches [32, 37, 38, 39, 40, 41, 42, 27, 25, 26, 43, 44] have extensively addressed these challenges but often necessitate high computational budget to achieve both strong image manipulations and faithful content preservation. While some of these techniques are applicable to the distilled models [25, 26, 27], they still dilute the primary advantage of distilled diffusion models: efficient inference.

One of the main ingredients of the iCD framework is how it operates with guided diffusion processes. Recently, dynamic guidance has been proposed to improve distribution diversity without noticeable loss in image quality [45, 46]. The key idea is to deactivate CFG for high diffusion noise levels to stimulate exploration at earlier sampling steps. In this work, we notice that dynamic CFG can facilitate image inversion while preserving the editability of the text-to-image diffusion models. Notably, dynamic CFG yields no computational overhead, entirely leveraging the efficiency gains from diffusion distillation. In our experiments, we demonstrate that invertible distilled models equipped with dynamic guidance are a highly effective inversion-based image editing tool.

To sum up, our contributions can be formulated as follows:

- We propose a generalized consistency distillation framework, invertible Consistency Distillation (iCD), enabling both high-fidelity text-to-image generation and accurate image encoding in around $3-4$ sampling steps.

- We investigate dynamic classifier-free guidance in the context of image inversion and text-guided editing. We demonstrate that it preserves editability of the text-to-image diffusion models while significantly increasing the inversion quality for free.

- We apply iCD to large-scale text-to-image models such Stable Diffusion 1.5 [4] and XL [1] and extensively evaluate them for image editing problems. According to automated and human studies, we confirm that iCD unlocks faithful text-guided image editing for $6-8$ steps and is comparable to state-of-the-art text-driven image manipulation methods while being multiple times faster.

## 2 Background

**Diffusion probabilistic models** DPMs [29, 28, 47] are a class of generative models producing samples from a simple, typically standard normal, distribution by solving the underlying Probability Flow ODE [30, 48], involving iterative *score function* estimation. DPMs are trained to approximate the score function and employ dedicated diffusion ODE solvers [49, 50, 48] for sampling. DDIM [49] is a simple yet effective solver, widely used in text-to-image models and operating in around 50 steps. A single DDIM step from $x_t$ to $x_s$ can be formulated as follows:

$$x_s^w = \text{DDIM}(x_t, t, s, c, w) = \sqrt{\frac{\alpha_s}{\alpha_t}} x_t + \epsilon_\theta^w(x_t, t, c)\left(\sqrt{1-\alpha_s} - \sqrt{\frac{\alpha_s}{\alpha_t}}\sqrt{1-\alpha_t}\right), \quad (1)$$

where $\alpha_s, \alpha_t$ are defined according to the diffusion schedule [47], and $\epsilon_\theta^w(x_t, t, c) = \epsilon_\theta(x_t, t, \oslash) + w(\epsilon_\theta(x_t, t, c) - \epsilon_\theta(x_t, t, \oslash))$ is a linear combination of conditional and unconditional noise predictions called as Classifier-Free Guidance (CFG) [51], used to improve the image quality and context alignment in conditional generation. In the following, we omit the condition $c$ for simplicity.

Due to reversibility, the PF ODE can be solved in both directions: encoding data into the noise space and decoding it back without additional optimization procedures. We refer to this process as *inversion* [32] where encoding and decoding correspond to *forward* and *reverse* processes, respectively.

**Consistency Distillation** CD [10, 12, 13] is the recent state-of-the-art diffusion distillation approach for few-step image generation, which learns to integrate the PF ODE induced with a pretrained diffusion model. In more detail, the model $f_\theta$ is trained to satisfy the self-consistency property:

$$\mathcal{L}_{\text{CD}}(\theta) = \mathbb{E}\left[d(f_\theta(x_{t_{n-1}}^w, t_{n-1}), f_\theta(x_{t_n}, t_n))\right] \to \min_\theta, \quad (2)$$

where $t_n \in \{t_0, ..., t_N\}$ is a discrete time step, $d(\cdot, \cdot)$ denotes a distance function and $x_{t_{n-1}}^w$ is obtained with a single step of the DDIM solver from $t_n$ to $t_{n-1}$ using the teacher diffusion model. The optimum of (2) is defined by the boundary condition, $f_\theta(x_{t_0}, t_0) = x_{t_0}$. Therefore, consistency models (CMs) learn the transition from any trajectory point to the starting one: $f_\theta(x_{t_n}, t_n) = x_{t_0}, \forall t_n \in \{t_0, ..., t_N\}$. Consequently, CMs imply a single step generation. However, approximating the entire trajectory using only one step remains highly challenging, leading to unsatisfactory results in practice. To address this, [10] proposes stochastic *multistep consistency sampling* that iteratively predicts $x_{t_0}$ using $f_\theta$ and goes back to the intermediate points using the forward diffusion process.

The competitive performance of consistency models has stimulated their rapid adoption for text-to-image generation [52, 18, 53]. Nevertheless, we believe that CMs have not yet fully realized their potential in downstream applications, where DPMs excel. One of the reasons is that, unlike DPMs, CMs do not support the inversion process. This work aims to unlock this ability for CMs.

**Dynamic guidance** State-of-the-art text-to-image models employ large CFG scales to achieve high image quality and textual alignment. However, it often leads to the reduced diversity of generated images. To address this, *dynamic classifier-free guidance* [45, 46, 54] has recently been proposed to improve distribution diversity without noticeable loss in generative performance. CADS [45] gradually increases the guidance scale from zero to the initial high value over the sampling process, Figure 2a. Alternatively, [46] proposes deactivating the guidance for low and high noise levels and using it only on the middle time step interval, Figure 2b. Both strategies suggest that the unguided process

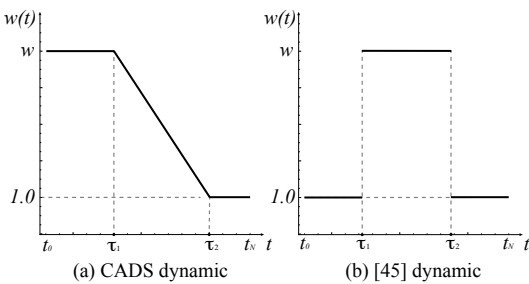

(a) CADS dynamic      (b) [45] dynamic

Figure 2: Dynamic CFG strategies.

at high noise levels is responsible for better distribution diversity without compromising sampling quality. In addition, the authors [46] demonstrate that guidance at low noise levels has a minor effect on the performance and can be omitted to avoid extra model evaluations for guidance calculation. Both dynamic techniques are controlled by two hyperparameters: $\tau_1$ and $\tau_2$, which are responsible for the value of dynamic CFG $w(t)$. In our work, we focus on the CADS formulation.

## 3 Method

This section introduces the invertible Consistency Distillation (iCD) framework, which comprises forward and reverse consistency models. First, we formulate the forward CD procedure that encodes images into latent noise. Then, we describe multi-boundary generalization of iCD to enable deterministic multistep inversion. Finally, we investigate the *dynamic guidance* technique from the inversion perspective.

### 3.1 Forward Consistency Distillation

Forward Consistency Distillation (fCD) works in the opposite way to CD. That is, it aims to map any point on the PF ODE trajectory to the latent noise (the last trajectory point).

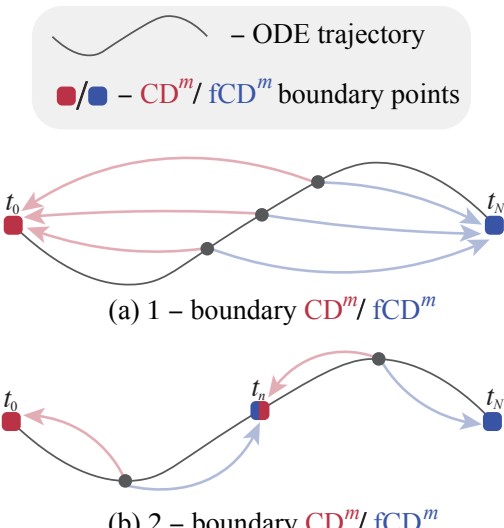

(a) 1 – boundary $\text{CD}^m$/ $\text{fCD}^m$

(b) 2 – boundary $\text{CD}^m$/ $\text{fCD}^m$

Figure 3: The proposed invertible Consistency Distillation framework consists of two models: the forward $m$-boundary model, $\text{fCD}^m$, and the reverse model, $\text{CD}^m$. (a) For $m = 1$, the reverse model corresponds to CD. More boundary points unlock the deterministic multistep inversion, e.g., (b) shows the case for $m = 2$.

The transition from CD to the forward counterpart is quite straightforward: the only thing that should be modified is the boundary condition. Precisely, the forward consistency model is constrained to be an identity function for the last trajectory point: $\boldsymbol{f_\theta}(\boldsymbol{x}_{t_N}, t_N) = \boldsymbol{x}_{t_N}$. Thus, fCD inherits the same consistency distillation loss (2) without incurring extra training costs. This way, the distilled model can transform any trajectory point to the last one: $\boldsymbol{f_\theta}(\boldsymbol{x}_{t_n}, t_n) = \boldsymbol{x}_{t_N}, \forall\, t_n$. To perform inversion, first, fCD encodes an image into noise and then CD decodes it back. The comparison between CD and fCD is shown in Figure 3a.

### 3.2 Multi-boundary Consistency Distillation

In practice, the encoding with fCD faces two challenges. Firstly, like in CD, a single-step prediction with fCD can be highly inaccurate. However, this cannot be easily addressed since the multistep consistency sampling [10] is not applicable to fCD. Concretely, intermediate points cannot be obtained from the latent noise using the forward diffusion process. Secondly, even if fCD is accurate, the multistep sampling is not suitable for decoding, as its stochastic nature prevents the reconstruction of real images. So, to improve the prediction accuracy of fCD and reduce the reconstruction error of CD, it is necessary to formulate a deterministic multistep procedure for both models.

Recent approaches [53, 13] generalize the CD framework to a multistep regime and allow approximating arbitrary trajectory intervals in the reverse direction. However, these methods focus solely on the generation quality, without supporting the inversion. Thus, inspired by these works, we propose a multi-boundary CD, that unlocks deterministic multistep inversion with the distilled models and carries similar training costs as the classical CD methods.

Specifically, we divide the solution interval, $\{t_0, ..., t_N\}$, into $m$ segments and perform the distillation on each of these segments separately. This way, we obtain a set of single-step consistency models operating on different intervals and boundary points. This formulation is valid for both CD and fCD and hence can enable deterministic multistep inversion. We provide an illustration of 2-boundary CD and fCD in Figure 3b. We denote the multi-boundary reverse and forward models as $\text{CD}^m$ and $\text{fCD}^m$.

Formally, we consider $\text{CD}^m$ and $\text{fCD}^m$ using the following parametrization, inspired by [53, 13].
$$\boldsymbol{x}_{s_t^m} = \boldsymbol{f_\theta}^m(\boldsymbol{x}_t, t, s_t^m, w) = \text{DDIM}(\boldsymbol{x}_t, t, s_t^m, w), \tag{3}$$
where $s_t^m$ is the boundary time step depending on the number of boundaries, $m$, and the current time step $t$. For instance, let $m = 1$, then $s_t^1 = t_0$ for $\text{CD}^1$ and $s_t^1 = t_N$ for $\text{fCD}^1$. Note that we learn a single model, the multistep sampling is achieved by varying $s_t^m$ during inference. The training objective remains the same as (2), avoiding additional training costs compared to CD. The only limitation is that the number of segments and the corresponding boundary time steps must be set before the training.

## 3.3 Training fCD$^m$ and CD$^m$

We train fCD$^m$ and CD$^m$ separately, initializing both with the same teacher model. We use the same loss with a difference only in boundary time steps. However, a notable difference is the CFG scale, $w$. For CD$^m$, we preliminarily embed the model on guidance, following [20], to use various $w$ during sampling and avoid extra model evaluations. For fCD$^m$, we consider an unguided model with a constant $w = 1$. The reason is that the guided encoding ($w > 1$) leads to out-of-distribution latent noise [32], and as a result, to poor image reconstruction. We confirm this intuition in Section 3.4. Finally, we find that $m=3-4$ provides competitive generation and inversion quality for large-scale text-to-image models [4, 1].

**Preservation losses** The procedure described above already provides decent inversion quality but still does not match the teacher inversion performance. To reduce the gap between them, we propose the forward and reverse preservation losses aimed at making CD$^m$ and fCD$^m$ more consistent with each other and improve the inversion accuracy. These losses can additionally be turned on during training. Below, we denote the parameters of CD$^m$, fCD$^m$ as $\boldsymbol{\theta}^+, \boldsymbol{\theta}^-$, respectively.

The forward preservation loss modifies only fCD$^m$ and is described as follows:

$$\mathcal{L}_{\mathrm{f}}(\boldsymbol{\theta}^-, \boldsymbol{\theta}^+) = \mathbb{E}\left[d\left(\boldsymbol{f}_{\boldsymbol{\theta}^-}^m(\boldsymbol{f}_{\boldsymbol{\theta}^+}^m(\boldsymbol{x}_{s_t^m})), \boldsymbol{x}_{s_t^m}\right)\right] \to \min_{\boldsymbol{\theta}^-}, \tag{4}$$

For simplicity, we omit some notation. In a nutshell, we sample a noisy image $\boldsymbol{x}_{s_t^m}$ for a boundary time step $s_t^m$, then make a prediction using CD$^m$ and force fCD$^m$ to predict the same $\boldsymbol{x}_{s_t^m}$. This approach encourages CD$^m$ and fCD$^m$ to be consistent with each other.

The reverse preservation loss provides the same intuition but with a difference in the optimized model (CD$^m$ instead of fCD$^m$) and prediction sequence. That is, we first make a prediction using fCD$^m$ and then use CD$^m$. We denote it as $\mathcal{L}_{\mathrm{r}}(\boldsymbol{\theta}^-, \boldsymbol{\theta}^+)$. In our experiments, we calculate the preservation losses only for the unguided reverse process ($w = 1$).

**Putting it all together** We present our final pipeline for the case where fCD$^m$ and CD$^m$ are trained jointly starting from the pretrained diffusion model. However, it is possible to learn them by one or take an already pretrained consistency model and learn the rest one. The final objective consists of two consistency losses with the proposed multi-boundary modification and two preservation losses:

$$\mathcal{L}_{\mathrm{iCD}}(\boldsymbol{\theta}^+, \boldsymbol{\theta}^-) = \mathcal{L}_{\mathrm{CD}}(\boldsymbol{\theta}^+) + \mathcal{L}_{\mathrm{CD}}(\boldsymbol{\theta}^-) + \lambda_{\mathrm{f}}\mathcal{L}_{\mathrm{f}}(\boldsymbol{\theta}^-, \boldsymbol{\theta}^+) + \lambda_{\mathrm{r}}\mathcal{L}_{\mathrm{r}}(\boldsymbol{\theta}^+, \boldsymbol{\theta}^-) \tag{5}$$

In this way, the proposed approach can compete with the state-of-the-art inversion methods using heavyweight diffusion models. We present technical details about the training in Appendix A.

## 3.4 Dynamic Classifier-Free Guidance Facilitates Inversion

As previously discussed, dynamic guidance [45, 46] provides promising results for both faithful and diverse text-to-image generation. In this work, we reveal that dynamic CFG is also an effective technique for improving inversion accuracy as shown in Figure 4b. Below, we delve into the questions when and why dynamic guidance might facilitate image inversion while preserving the generative performance. To answer these questions, we conduct experiments using Stable Diffusion 1.5 with DDIM solver for 50 steps and maximum CFG scale set to 8.0.

**Dynamic guidance for decoding** We start with the dynamic CFG analysis at the decoding stage using the unguided encoding process following the prior work [32]. First, we wonder at which time steps the guidance has the most significant impact on reconstruction quality. To this end, we evaluate MSE between real and reconstructed images for different CFG turn-on thresholds $\mathbf{T}$. If $t > \mathbf{T}$, we set $w = 1.0$, otherwise, the CFG scale is set to its initial value 8.0. In Figure 4a, we observe an exponential decrease in reconstruction error, implying that the absence of CFG at higher noise levels is essential for achieving more accurate inversion. Figure 4b confirms this intuition qualitatively. These results are consistent with [45, 46], which also suggest turning off the guidance at high noise levels but motivating this from the perspective of diversity improvement.

Then, we investigate the influence of various $\tau_1, \tau_2$ from the CADS dynamic (Figure 2a) on the inversion and generation performance. We aim to identify an operating point providing both strong

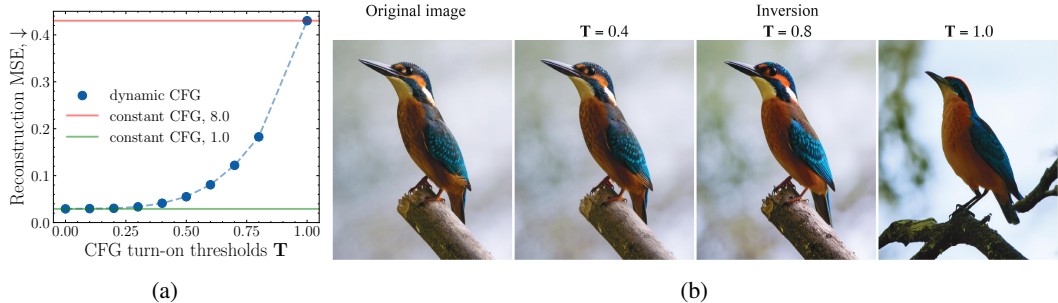

(a)            (b)

Figure 4: (a) Reconstruction error of the decoding process for different CFG turn-on thresholds. (b) Image inversion examples for different CFG turn-on thresholds $\mathbf{T}$. Guidance at high noise levels ($\mathbf{T} = 1.0$) drastically degrades the inversion quality.

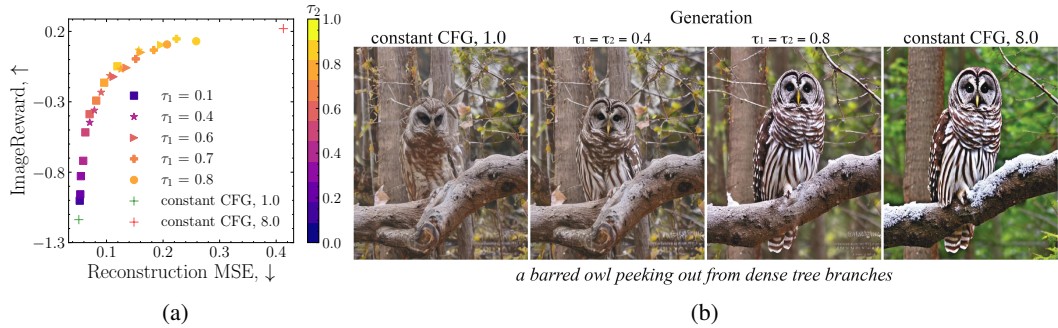

(a)            (b)

Figure 5: (a) Trade-off between generation performance (IR) and reconstruction quality (MSE) provided by different $\tau_1, \tau_2$. (b) Generation examples for dynamic and constant CFG scales. The points around $\tau_1 = \tau_2 = 0.8$ provide preferable trade-off between generation and inversion performance.

|  | Enc, No CFG | Enc, d.CFG, $\tau$=0.6 | Enc, d.CFG, $\tau$=0.8 | Enc, CFG |
|---|---|---|---|---|
| Dec. No CFG | 11.0 | 36.6 | 63.4 | 100.5 |
| Dec. d.CFG $\tau$=0.6 | 11.4 | 23.5 | 67.8 | 102.6 |
| Dec. d.CFG $\tau$=0.8 | 15.0 | 14.6 | 52.2 | 108.5 |
| Dec. CFG | 19.0 | 19.2 | 19.0 | 102.1 |
| Latent NLL, $\downarrow$ | 1.401 | 1.409 | 1.415 | 1.428 |

Table 1: FID-5k for SD1.5 starting from the noise latents obtained using different encoding strategies, and NLL for these latents. Though encoding with dynamic CFG produces consistently more plausible latents than constant CFG, the unguided encoding remains preferable.

generation performance and faithful image inversion. Thus, we evaluate generation performance using the ImageReward [55] (IR) on top of randomly generated samples for $1000$ COCO2014 prompts [56]. The inversion accuracy is estimated in terms of MSE between original and reconstructed samples. Figure 5a presents the results for varying $\tau_1$ and $\tau_2$. It can be seen that several points for $\tau_1 \geq 0.7$ offer slightly lower text-to-image performance but exhibit significantly better reconstruction quality compared to the constant CFG scale, $8.0$. Moreover, we notice that the settings where $\tau_1 = \tau_2$ perform similarly to those where $\tau_1 < \tau_2$. Consequently, in all our experiments, we consider a single $\tau$ representing the case where $\tau_1 = \tau_2$ and use $\tau = 0.7$ and $\tau = 0.8$. This means that $CD^m$ follows unguided sampling for $t > \tau$ and sets the initial CFG scale for $t \leq \tau$.

Note that the setting with $\tau=\tau_1=\tau_2$ corresponds to a step CFG function $w(t)$, which yields a distinct advantage for distilled models. The linearly changing CFG scales are not applicable to the processes with large discretization steps, typical for distilled diffusion models. Therefore, such a CFG schedule needs to be distilled into the model during training, making it less flexible for different generation and editing settings. In contrast, the step CFG function enables dynamic CFG for already pretrained distilled models, operating with different constant CFG scales.

| Ours, 4 steps | Teacher, 50 steps | Ours, 4 steps | Teacher, 50 steps |

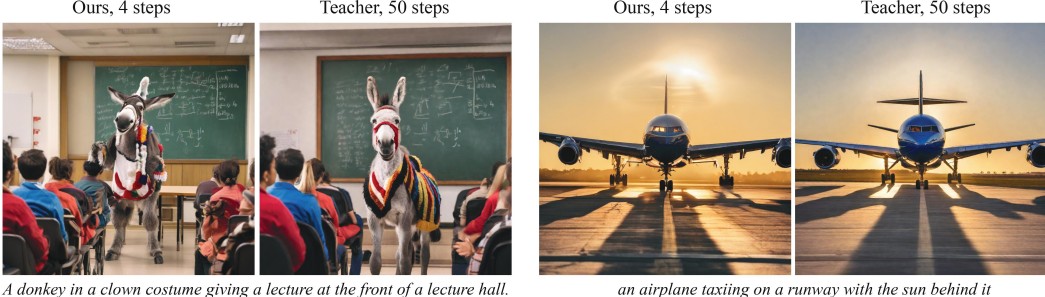

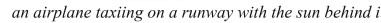

*A donkey in a clown costume giving a lecture at the front of a lecture hall. The blackboard has mathematical equations on it.*

*an airplane taxiing on a runway with the sun behind it*

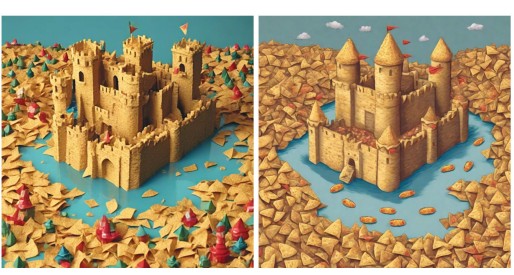

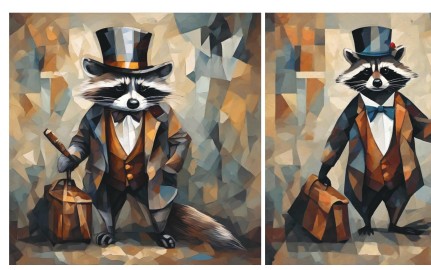

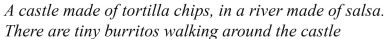

*A castle made of tortilla chips, in a river made of salsa. There are tiny burritos walking around the castle*

*A raccoon wearing formal clothes, wearing a tophat and holding a cane. The raccoon is holding a garbage bag. Oil painting in the style of cubism.*

Figure 6: Few examples of text-to-image generation using the iCD-XL model for 4 steps.

**Dynamic guidance for encoding** Next, we investigate the guidance role in tackling the encoding problem. In Table 1, we compare noise latents encoded using various guidance strategies. The quality of the noise latents is estimated by evaluating the generation performance starting from these latents with a fixed CFG scale of $8.0$. As the performance measure, we calculate FID for $5000$ image-text pairs from COCO [56].

We observe that the latents obtained with a consistently high CFG scale exhibit the worst generative performance, indicating their out-of-domain nature. While dynamic guidance produces significantly more plausible latents, it still falls short of the unguided encoding in most cases. To further validate these results, we estimate the negative log-likelihood (NLL) of the encoded latents under different CFG settings in Table 1 (Bottom). NLL is calculated with respect to the standard normal distribution. While NLL decreases for dynamic CFG with lower $\tau$, the encoding without guidance ($w$=1) provides the highest likelihood value. Therefore, in all our experiments, we maintain $w$=1 for the encoding and train the forward distilled models (fCD) on the unguided teacher process.

# 4 Experiments

In the following experiments, we apply our approach to text-to-image diffusion models of different scales: SD1.5 [57] and SDXL [1], and denote them iCD-SD1.5 and iCD-XL, respectively. We provide the training details in Appendix A.

Initially, we illustrate the inversion capability of the proposed framework. Then, we consider the text-guided image editing problem and demonstrate that our approach outperforms or is comparable to significantly more expensive baselines.

Before diving into the main experiments, we present a few generated samples using iCD-XL for 4 steps in Figure 6. Additional quantitative and qualitative results are provided in Appendix B. The results confirm that the distilled model demonstrates strong text-to-image generation performance.

## 4.1 Inversion quality of iCD

Here, we analyze the reconstruction capabilities of iCD-SD1.5 under various configurations. Specifically, we explore the contribution of the different pipeline components, such as the number of steps, preservation losses, and dynamic CFG, to inversion performance. Our forward model is run without

| Configuration | LPIPS ↓ | DinoV2 ↑ | PSNR ↑ |
|---|---|---|---|
| **Unguided decoding setting** | | | |
| fDDIM$^{50}$ \| DDIM$^{50}$ | 0.167 | 0.834 | 22.98 |
| **A** fCD$^2$ \| CD$^2$ | 0.332 | 0.632 | 17.75 |
| **B** fCD$^3$ \| CD$^3$ | 0.317 | 0.649 | 18.42 |
| **C** fCD$^4$ \| CD$^4$ | 0.276 | 0.715 | 19.19 |
| **E** ~~fCD$^4$~~ + $\mathcal{L}_\mathrm{f}$ \| CD$^4$ | 0.484 | 0.554 | 16.40 |
| **F** fCD$^4$ + $\mathcal{L}_\mathrm{f}$ \| CD$^4$ | 0.248 | 0.728 | 20.01 |
| **G** fCD$^4$ + $\mathcal{L}_\mathrm{f}$ \| CD$^4$ + $\mathcal{L}_\mathrm{r}$ | **0.198** | **0.837** | **22.27** |
| **Guided decoding setting**, $w = 8$ | | | |
| fDDIM$^{50}$ \| DDIM$^{50}$ | 0.479 | 0.534 | 14.12 |
| fDDIM$^{50}$ \| DDIM$^{50}$ + d.CFG | 0.279 | 0.726 | 19.58 |
| **H** fCD$^4$ \| CD$^4$ | 0.476 | 0.550 | 13.87 |
| **I** fCD$^4$ \| CD$^4$ + d.CFG | 0.370 | 0.650 | 16.72 |
| **J** fCD$^4$ + $\mathcal{L}_\mathrm{f}$ \| CD$^4$ + d.CFG | 0.317 | 0.698 | 17.98 |
| **K** fCD$^4$ + $\mathcal{L}_\mathrm{f}$ \| CD$^4$ + $\mathcal{L}_\mathrm{r}$ + d.CFG | **0.273** | **0.749** | **19.66** |

Table 2: Exploration of iCD-SD1.5 configurations in terms of image inversion performance.

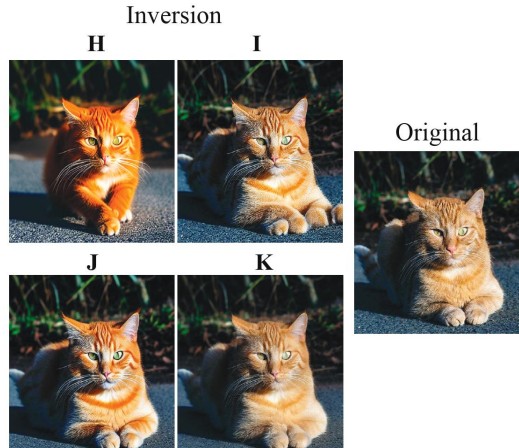

Figure 7: Influence of the dynamic guidance and preservation losses on image inversion with iCD.

CFG ($w = 1$), while for the reverse model, we consider two settings: unguided ($w = 1$) and guided ($w = 8$), both of which are important in practice.

**Configuration** To evaluate the inversion quality, we consider 5K images and the corresponding prompts from the MS-COCO dataset [56]. We measure the reconstruction quality using LPIPS [58], PSNR and cosine distance in the DinoV2 [59] feature space. As for the reference, the teacher inversion with a disabled CFG scale is considered. For the dynamic guidance, we use $\tau = 0.7$. The coefficients for the preservation losses are equal to $\lambda_\mathrm{f} = 1.5$ and $\lambda_\mathrm{r} = 1.5$.

**Results** The results are presented in Table 2. First, configurations (**A**-**C**) evaluate the number of the forward and inverse models inference steps. We observe that the reconstruction quality improves as the number of steps increases. In our main experiments, we consider 3 and 4 steps.

Then, (**E**-**G**) examine the preservation losses. In (**E**), we learn the forward model in the encoder [60] regime using the forward preservation loss only. This experiment reveals that the consistency loss contributes significantly to inversion performance. (**F**, **G**) show that both losses improve the inversion, with the latter approaching the quality of the teacher model.

Finally, we explore the dynamic CFG and preservation losses under the guided decoding setting (**I**-**K**) and compare them to the setting (**H**), which does not employ any boosting techniques. From the configurations (**I**, **J**, **K**), we can see that all techniques provide significant contribution to the reconstruction quality. In Figure 7, we visualize their influence on inversion. It can be seen that the dynamic CFG (**I**) is rather responsible for global object preservation, while the preservation losses (**J**,**K**) rather improve fine-grained details. We note that the final configuration (**K**) provides comparable inversion quality to the unguided process while preserving the editing capabilities due to the activated guidance. More visual examples of inversion and quantitative results are in Appendix C.

## 4.2 Text-guided image editing

In this section, we apply the proposed iCD to the text-guided image editing problem. For the SD1.5 model, we use the Prompt-to-Prompt (P2P) method [61]. We vary two hyperparameters: the cross-attention and self-attention steps balancing between editing strength and preservation of the reference image. We also apply our approach to MasaCTRL [62] in Appendix D. For the SDXL model, we follow the ReNoise [25] evaluation setting and just change the source prompt during decoding according to [63].

**Metrics** We measure editing performance using both automatic metrics and human-study. The former uses two metrics: 1) to estimate the preservation of the reference image, we calculate the cosine distance between images in the DinoV2 feature space; 2) as an editing quality measure, we use the CLIP score between the edited image and the target prompt. For human evaluation, we employ professional assessors who successfully completed assessment tasks. We show them the source and target prompts, reference image and two images produced with the methods under the comparison

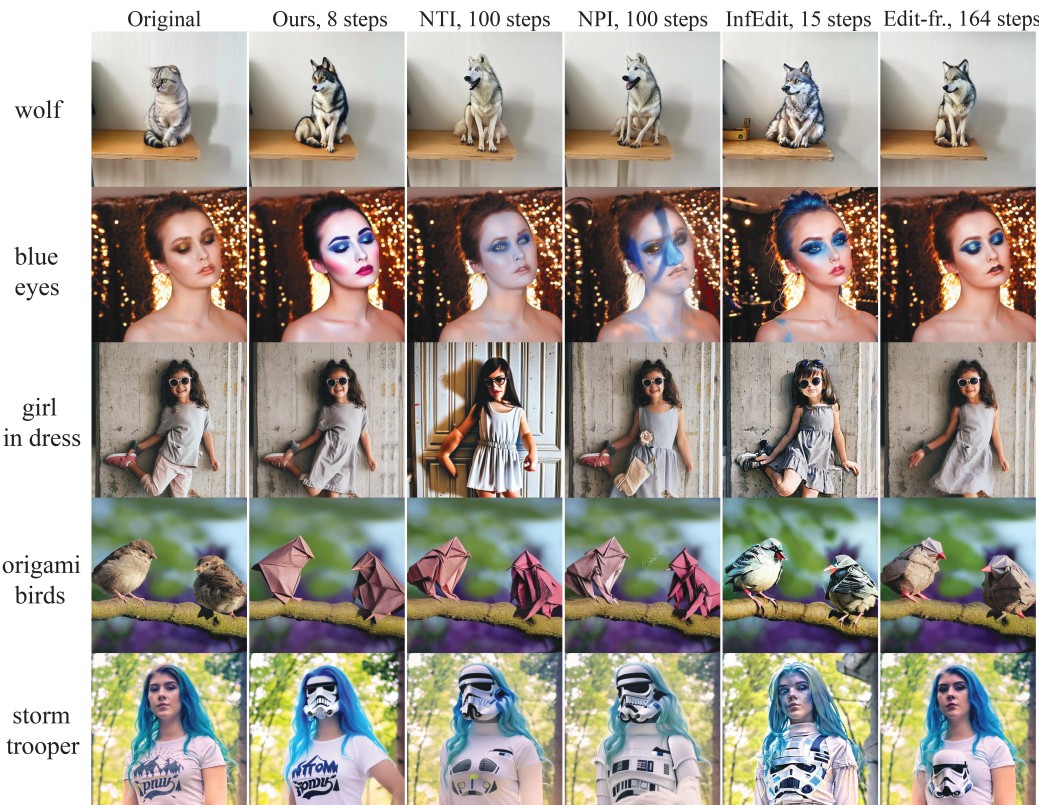

|  | Original | Ours, 8 steps | NTI, 100 steps | NPI, 100 steps | InfEdit, 15 steps | Edit-fr., 164 steps |
|--|----------|---------------|----------------|----------------|-------------------|---------------------|

Figure 8: Image editing examples produced by our method (iCD-SD1.5) and the baseline approaches.

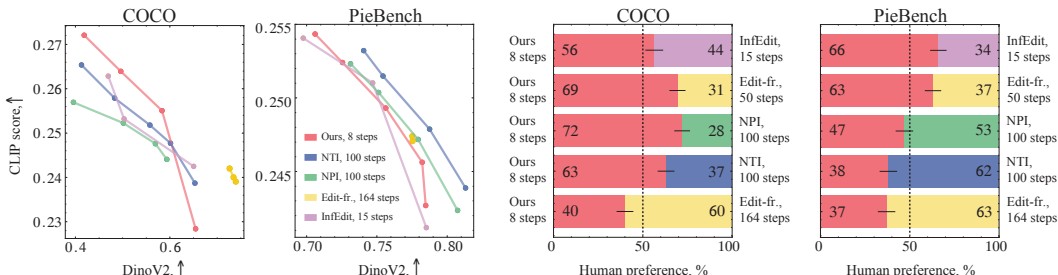

Figure 9: Quantitative comparisons between different editing approaches based on SD1.5: automatic metrics (left) and human preference study (right).

and ask the question: *which of the edited images do you prefer more taking into account the editing strength and reference preservation?*

**Benchmarks** In our experiments, we consider two benchmarks: PieBench [31] and a manually created COCO evaluation set based on text-paired images from the MS-COCO dataset [56]. PieBench [31] consists of various types of editing, for example, replacement, addition, or deletion. We take 420 examples of realistic images, including all types of editing. COCO focuses solely on the replacement task as one of the most popular among practitioners. This benchmark contains 140 image-text pairs.

### 4.2.1 Text-guided image editing with iCD-SD1.5

**Configuration** To provide the editing with the SD1.5 model, we consider iCD using 4 forward and 4 reverse steps trained with both preservation losses ($\lambda_f = 1.5$, $\lambda_r = 1.5$). In Appendix D, we also present the results for iCD using 3 steps, which is not much worse than the 4 step model. We set the hyperparameters of the dynamic CFG to $\tau = 0.8$ and maximum CFG scale to 19.0.

| Automatic evaluation | | | |
|---|---|---|---|
| Configuration | CLIP score, T ↑ | DinoV2 ↑ | CLIP score, I ↑ |
| COCO benchmark | | | |
| Ours, 8 | 0.267 | **0.472** | **0.748** |
| ReNoise Turbo, 44 | 0.265 | 0.399 | 0.705 |
| ReNoise SDXL, 150 | 0.227 | 0.431 | 0.732 |
| ReNoise LCM, 35 | 0.218 | 0.350 | 0.663 |
| PieBench | | | |
| Ours, 8 | 0.254 | **0.707** | **0.858** |
| ReNoise Turbo, 44 | 0.254 | 0.615 | 0.820 |
| ReNoise SDXL, 150 | 0.234 | 0.642 | 0.835 |
| ReNoise LCM, 35 | 0.236 | 0.736 | 0.865 |

| Human preference, % | | | |
|---|---|---|---|
| COCO benchmark | | PieBench | |
| Ours, 8 | **64 ± 4.9** | Ours, 8 | **63 ± 5.1** |
| ReNoise Turbo, 44 | 36 ± 4.9 | ReNoise Turbo, 44 | 37 ± 5.1 |
| Ours, 8 | **91 ± 2.8** | Ours, 8 | **78 ± 4.3** |
| ReNoise SDXL, 150 | 9 ± 2.8 | ReNoise SDXL, 150 | 22 ± 4.3 |
| Ours, 8 | **95 ± 1.8** | Ours, 8 | **76 ± 4.2** |
| ReNoise LCM, 35 | 5 ± 1.8 | ReNoise LCM, 35 | 24 ± 4.2 |

Table 3: Automatic metrics (top) and human evaluation (bottom) for iCD-XL and ReNoise [25].

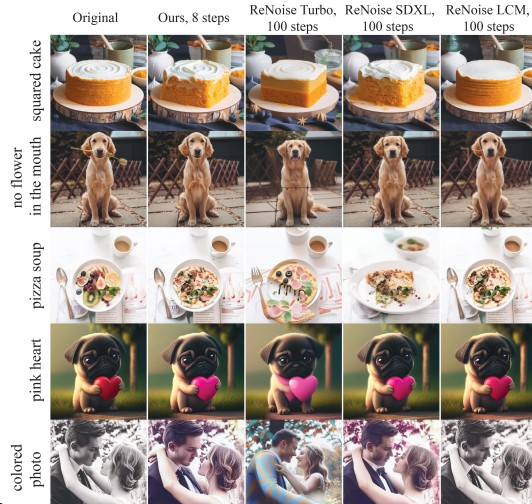

Figure 10: Editing examples using XL models.

**Baselines** We compare our approach with four baseline approaches, which provide state-of-the-art editing performance: Null-text Inversion (NTI), Negative-prompt Inversion (NPI) [32], InfEdit [26] and Edit-friendly DDPM [43]. All methods except InfEdit are diffusion-based approaches using more than 100 steps. Moreover, NTI employs the additional high-cost optimization procedure to improve inversion quality. InfEdit operates with the distilled diffusion model, latent consistency distillation [18], but utilizing virtual inversion. All methods use P2P and we vary all possible hyperparameter values to find the configurations that provide the best editing-preservation trade-off.

**Results** Figure 9 provides quantitative results for both benchmarks. We observe that the proposed iCD is comparable to the baseline approaches in most cases. Moreover, sometimes it can even outperform them while being multiple times faster. For instance, according to human preference on the COCO benchmark, our approach surpasses the InfEdit, NPI, NTI and Edit-friendly DDPM (50 steps). On the PieBench, it outperforms the InfEdit and Edit-friendly DDPM (50 steps). In addition, we provide qualitative results in Figure 8, which confirm the competitiveness of the proposed method. Additional visual examples can be found in Appendix D.

Notably, the performance of the proposed method is weaker on the PieBench benchmark compared to COCO, according to both automatic and human-based metrics. We attribute this to the increased complexity of editing tasks, which probably require more steps.

### 4.2.2 Text-guided image editing with iCD-XL

**Configuration** We consider the configuration using 4 steps, $\tau = 0.7$ and CFG scale equals 8.0.

**Baselines** We compare our approach to the recently proposed ReNoise [25], accurately following its guidelines. This method works with both distilled models (LCM-SDXL [18], SDXL-Turbo [19]) and original diffusion model, SDXL [1]. However, even for the distilled models, a significant number of steps is required to achieve decent performance.

**Results** The quantitative and qualitative comparisons are presented in Table 3 and Figure 10, respectively. According to the human evaluation, iCD-XL outperforms all ReNoise configurations. Based on the automatic evaluation, our approach provides better reference preservation (DinoV2 and CLIP score, I) while maintaining strong editing capabilities, as indicated by the CLIP score (T). We provide more visual examples in Appendix D.

## 5  Conclusion

The paper proposes a generalized consistency distillation framework that enables both accurate image inversion and solid generation performance using a few inference steps. Accompanied by the recently proposed dynamic guidance, the distilled models demonstrate highly efficient and accurate image manipulations, making a significant step towards real-time text-driven image editing.

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

# A  Training details of iCD

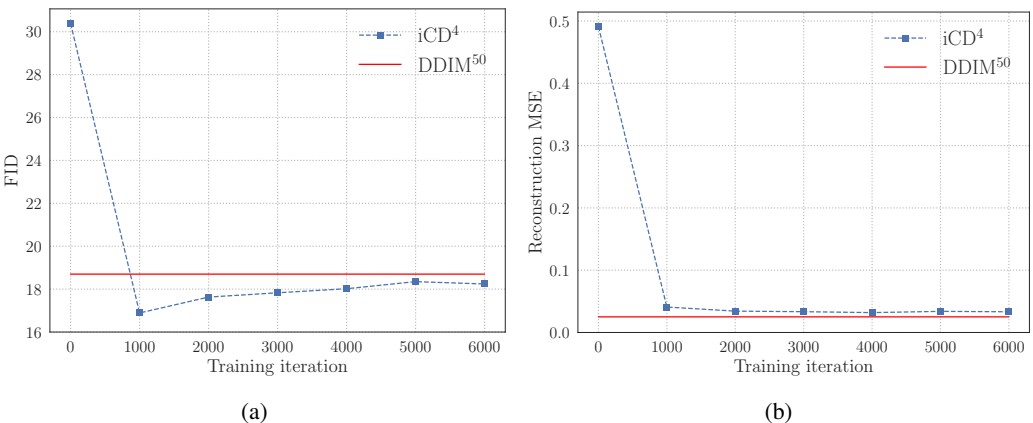

(a)  (b)

Figure 11: Training dynamics of iCD-SD1.5 in terms of FID (a) and reconstruction loss (b).

First, following [20], we preliminary distill classifier-free guidance using a conditional embedding added to the time step embedding. The goal is to use different $w$ and apply dynamic techniques during inference after consistency distillation. We perform CFG distillation for the following guidance scales: $1, 8, 12, 16, 20$ for SD1.5 and $1, 4, 6, 9, 10, 12, 13, 16, 18, 20$ for SDXL. At this stage, the model successfully approximates the guided teacher without hurting its performance.

Then, we perform multi-boundary consistency distillation using LoRA adapters with a rank of $64$ following [52]. We train forward and reverse adapters in parallel starting from the same teacher: CFG distilled SD1.5 or SDXL. For the SD1.5 model, we use a global batch size of $512$, and for the SDXL $128$. All models converge relatively fast, requiring about 6K iterations with a learning rate $8e{-}6$.

We find that the forward and reverse models provide promising generation and inversion quality for 3 or 4 steps. The regularization coefficients for the forward and reverse preservation losses are $\lambda_f{=}1.5$ and $\lambda_r{=}1.5$, respectively.

The iCD-SD1.5 models are trained for $\sim$36h and the iCD-XL ones for $\sim$68h on 8 NVIDIA A100 GPUs. We present the training dynamics in terms of FID and reconstruction MSE for iCD-SD1.5 in Figure 11.

For SD1.5 distillation, we use a $\sim$20$M$ subset of LAION 2B, roughly filtered using CLIP score [64]. For SDXL, we collect $\sim$7$M$ images with resolution $\geq 1024$, also curated to avoid poorly aligned text-image pairs and low quality images.

We set the following time steps for our configurations:

- 4 steps, $\tau = 0.8$: reverse model $[259, 519, 779, 999]$; forward model $[19, 259, 519, 779]$;
- 4 steps, $\tau = 0.7$: reverse model $[259, 519, 699, 999]$; forward model $[19, 259, 519, 699]$;
- 3 steps $\tau = 0.7$: reverse model $[339, 699, 999]$; forward model $[19, 339, 699]$;

# B  Image generation with iCD

We provide the generation performance of our distilled model in Table 4. The dynamic CFG ($\tau = 0.8$, $\tau = 0.7$) degrades in terms of ImageReward, while improving FID due to the increased diversity [45, 46]. The visual examples are presented in Figures 16, 17.

# C  Image inversion

In Table 5, we provide comparisons between the 3- and 4-step configurations of iCD-SD1.5, which perform similarly. Figure 14 shows the image inversions provided by our approach, NTI and NPI.

| Configuration | FID | CLIP score | ImageReward |
|---|---|---|---|
| DDIM$^{50}$ | 18.73 | 0.266 | 0.201 |
| CD$^4$ | 18.45 | 0.259 | 0.044 |
| CD$^3$ | 17.96 | 0.258 | $-0.009$ |
| CD$^4$ + d.CFG, $\tau = 0.8$ | 16.25 | 0.254 | $-0.193$ |
| CD$^4$ + d.CFG, $\tau = 0.7$ | 16.38 | 0.253 | $-0.217$ |
| CD$^3$ + d.CFG, $\tau = 0.7$ | 17.66 | 0.251 | $-0.351$ |

Table 4: Text-to-image performance of the SD1.5 model in terms of FID-5K, CLIP score and ImageReward for $w = 8$ using 5K prompts from the MS-COCO dataset.

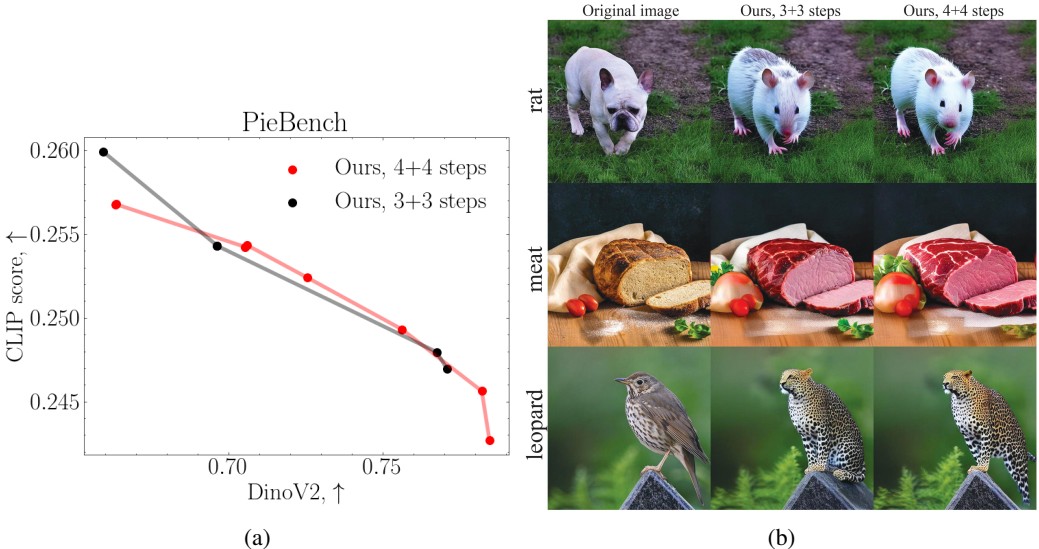

(a)

(b)

Figure 12: Quantitative (a) and qualitative (b) editing results using 3- and 4-step iCD-SD1.5 configurations on PieBench.

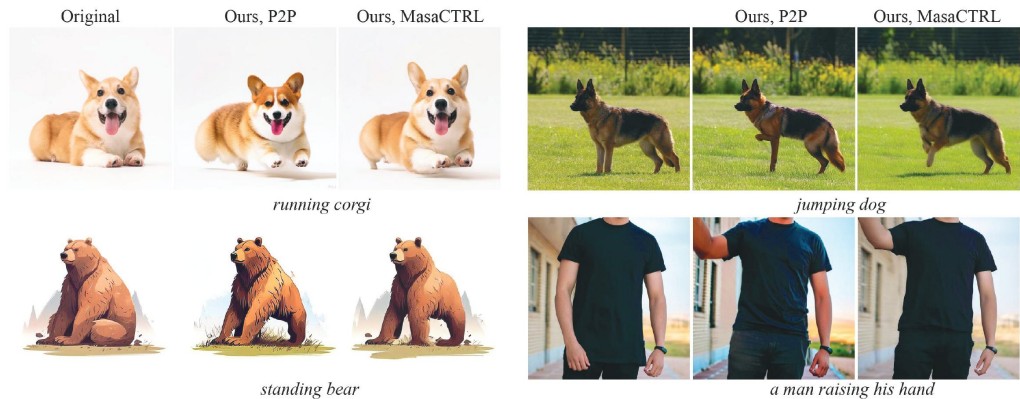

Figure 13: Comparison between Prompt2Prompt and MasaCTRL applied to our approach

Figure 15 shows the image inversions compared to the ReNoise method. In Table 6, we present the time required to invert a single image for different methods.

| Configuration | LPIPS $\downarrow$ | DinoV2 $\uparrow$ | PSNR $\uparrow$ |
|---|---|---|---|
| **Unguided decoding setting** | | | |
| fDDIM$^{50}$ \| DDIM$^{50}$ | 0.167 | 0.834 | 22.98 |
| **E**   fCD$^4 + \mathcal{L}_\mathrm{f}$ \| CD$^4 + \mathcal{L}_\mathrm{r}$ | 0.198 | 0.837 | 22.27 |
| **E**   fCD$^3 + \mathcal{L}_\mathrm{f}$ \| CD$^3 + \mathcal{L}_\mathrm{r}$ | 0.192 | 0.841 | 22.21 |
| **Guided decoding setting**, $w = 8$ | | | |
| fDDIM$^{50}$ \| DDIM$^{50}$ | 0.479 | 0.534 | 14.12 |
| fDDIM$^{50}$ \| DDIM$^{50}$ + d.CFG | 0.279 | 0.726 | 19.58 |
| **E**   fCD$^4 + \mathcal{L}_\mathrm{f}$ \| CD$^4 + \mathcal{L}_\mathrm{r}$ + d.CFG | 0.273 | 0.749 | 19.66 |
| **E**   fCD$^3 + \mathcal{L}_\mathrm{f}$ \| CD$^3 + \mathcal{L}_\mathrm{r}$ + d.CFG | 0.263 | 0.770 | 19.76 |

Table 5: More image inversion results for 3- and 4-step iCD-SD1.5. Dynamic CFG uses $\tau = 0.7$.

| Method | Ours 8 steps, SD1.5 | NTI, SD1.5 | NPI, SD1.5 | Ours 8 steps, SDXL | ReNoise, LCM-XL |
|---|---|---|---|---|---|
| Time, secs | $0.959 \pm .005$ | $116.4 \pm .1$ | $9.95 \pm .03$ | $1.56 \pm .07$ | $6.75 \pm .52$ |

Table 6: The time required to invert a single image for different approaches.

## D   Text-guided image editing

Figure 12 compares two iCD configurations (3 and 4 steps). We observe that both configurations perform similarly, with a slight preference for the 4-step configuration. We present additional visual results on image editing using the iCD-SD1.5 model in Figure 18 and the iCD-XL model in Figure 19.

Figure 13 provides a few examples of our approach combined with MasaCTRL [62] for non-rigid editing. The results demonstrate that our method can be used with various editing methods.

We also present the comparison with SDXL-Turbo using SDEdit [63] in Figure 20 (upper). We observe that SDXL Turbo significantly hurts reference image preservation due to stochasticity. This highlights the importance of accurate image inversion for editing. Moreover, we compare our approach with Intruct-Pix2Pix [7], Figure 20 (bottom), and observe that it outperforms Intruct-Pix2Pix in terms of both content preservation and editing strength while being training-free.

We present the annotation interface for the assessors in Figure 22. We calculate the confidence interval using bootstrap methods, splitting the human votes into $1,000$ subsets and then averaging the results and calculating the standard deviation.

## E   Failure cases

We present some inversion failure cases in Figure 21. Our method sometimes oversaturates images for high guidance scales and struggles to reconstruct complex details like human faces and hands

## F   Limitations

The iCD limitations include the requirement to predetermine boundary time steps before distillation, which may restrict the model flexibility. Additionally, the extra preservation losses necessitate increased computational resources during training. Furthermore, the editing method is susceptible to hyperparameter values, leading to inconsistent performance across different prompts. Finally, the quality of the current distillation techniques itself requires significant improvement, as it does not consistently achieve high standards across various scenarios.

# G Broader impacts

Our work can significantly enhance tools for artists, designers, and content creators, allowing for more precise and efficient manipulation of images based on textual inputs. This can democratize high-quality digital art creation, making it accessible to those without extensive technical skills. On the other hand, the ability to edit images easily and realistically can be misused to create misleading information or fake images, which can be particularly harmful and potentially influence public opinion and elections.

Original image          Ours, 8 steps          NPI, 100 steps          NTI, 100 steps

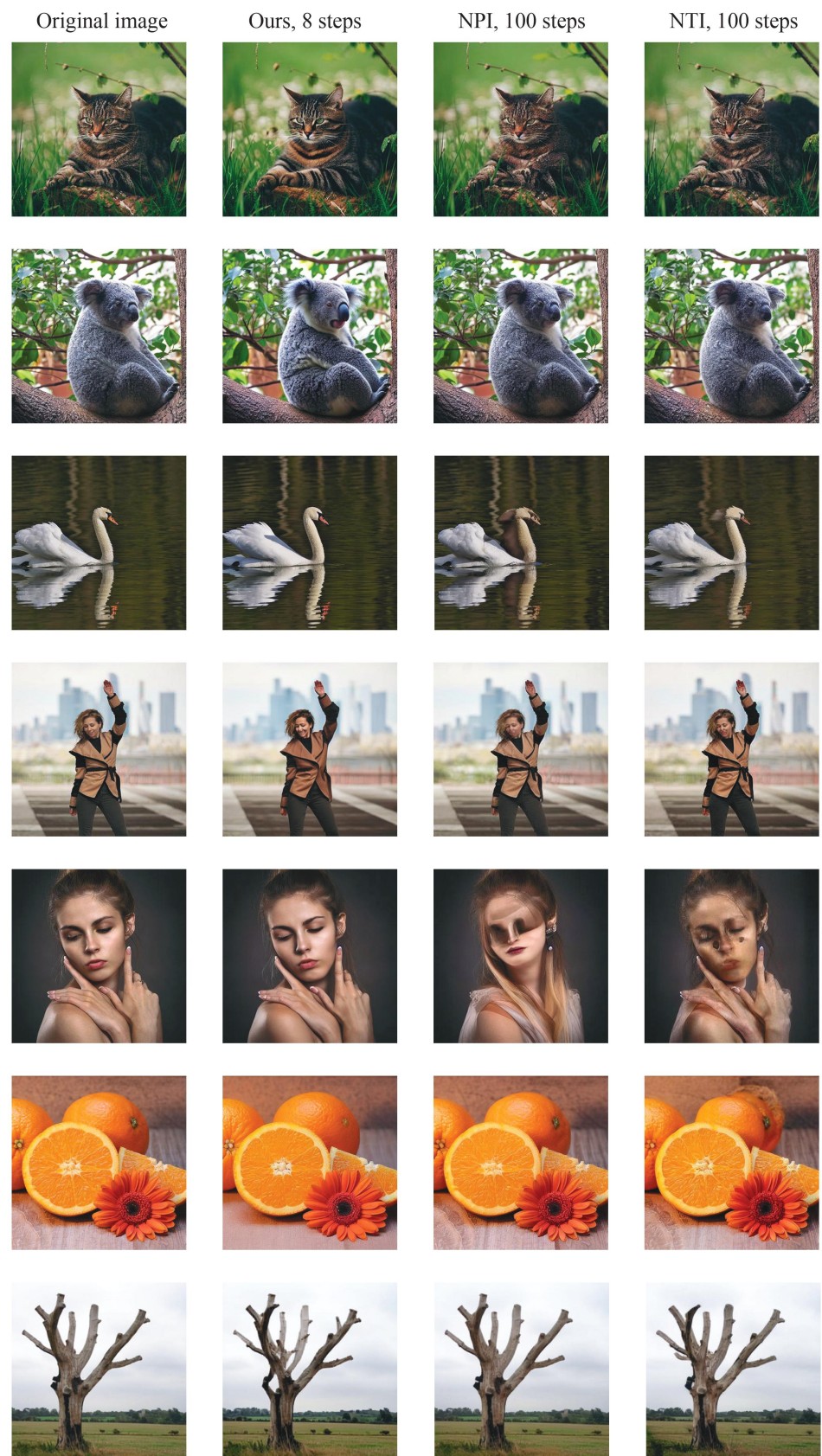

Figure 14: Inversion examples produced by the SD1.5-based models.

| Original image | Ours, 8 steps | ReNoise Turbo 44 steps | ReNoise LCM 35 steps | ReNoise SDXL 150 steps |

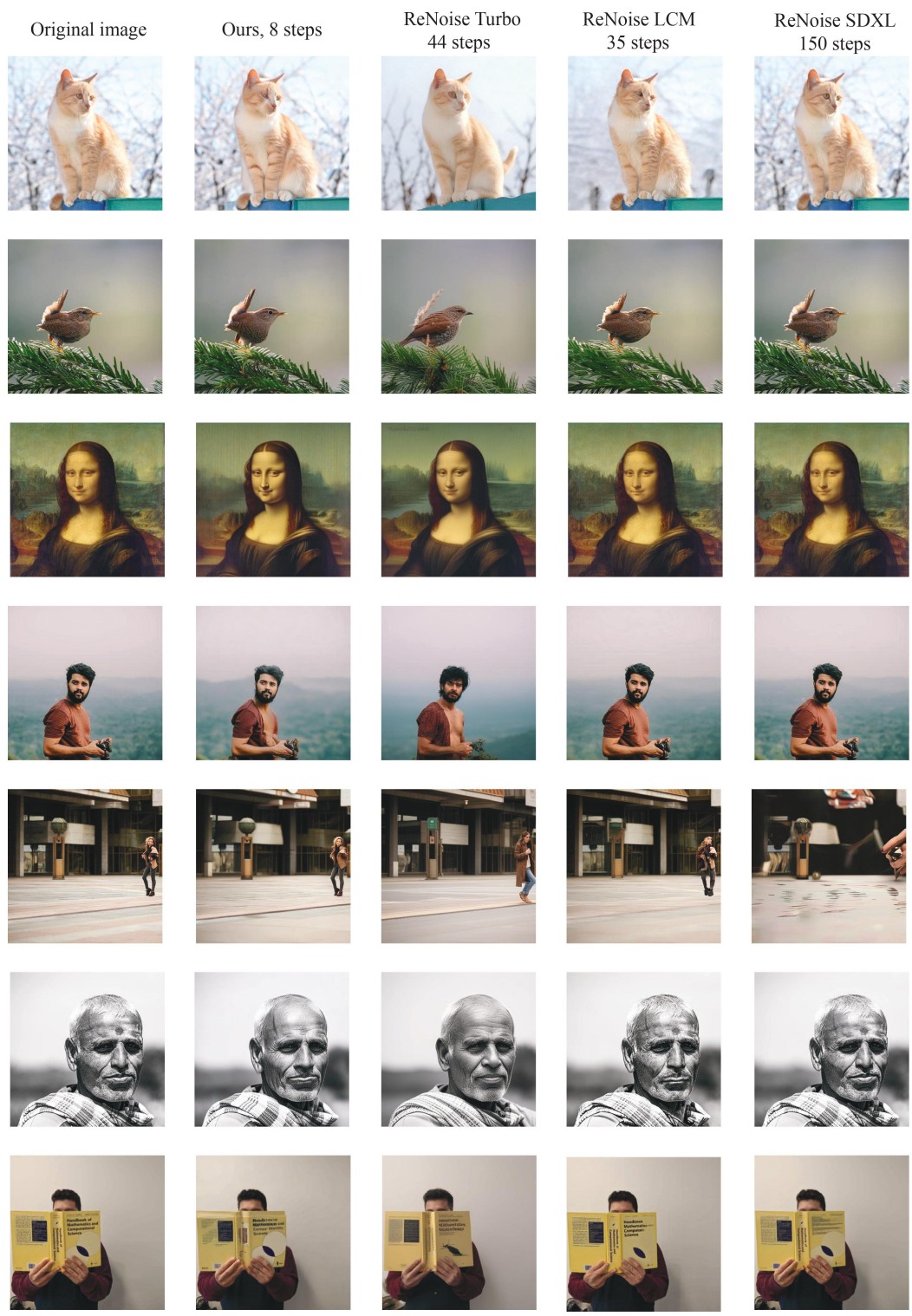

Figure 15: Inversion examples produced by the SDXL-based models.

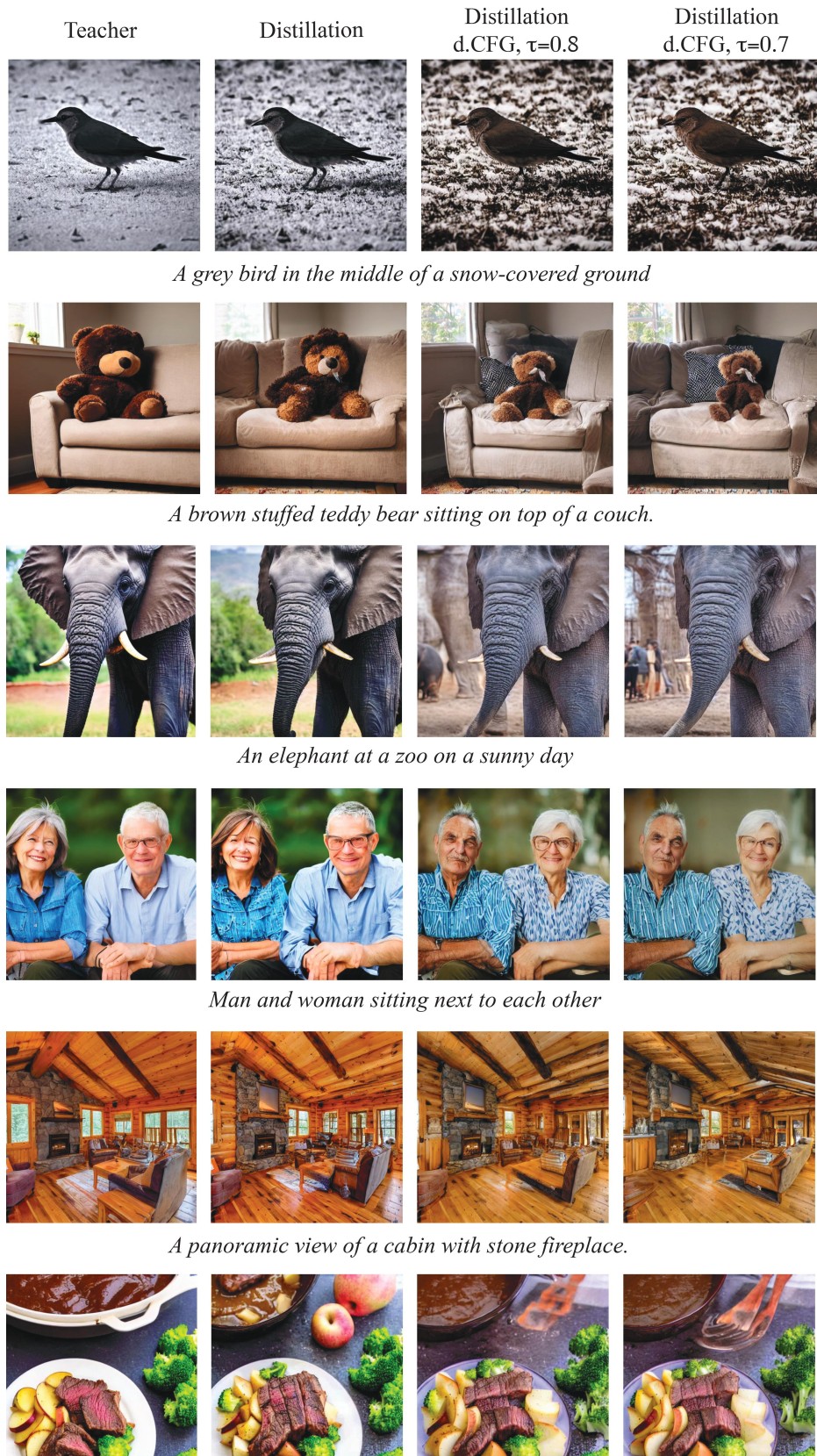

Figure 16: Generation examples using SD1.5, the proposed distilled method using 4 steps and dynamic CFG.

|   Teacher   |   Distillation   |   Distillation
d.CFG, τ=0.8   |   Distillation
d.CFG, τ=0.7   |

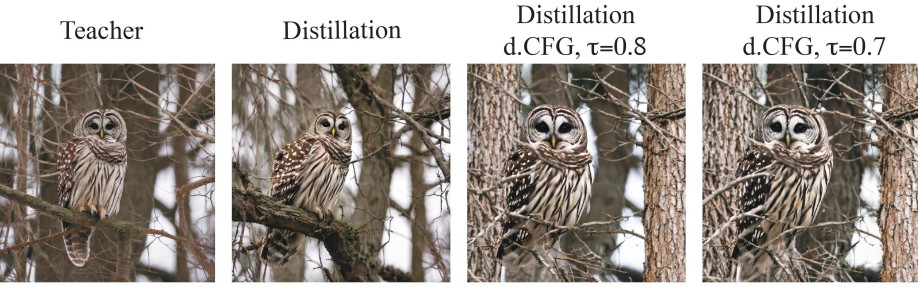

*a barred owl peeking out from dense tree branches*

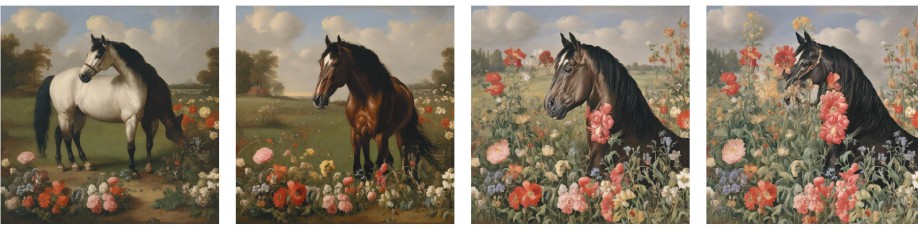

*a dutch baroque painting of a horse in a field of flowers*

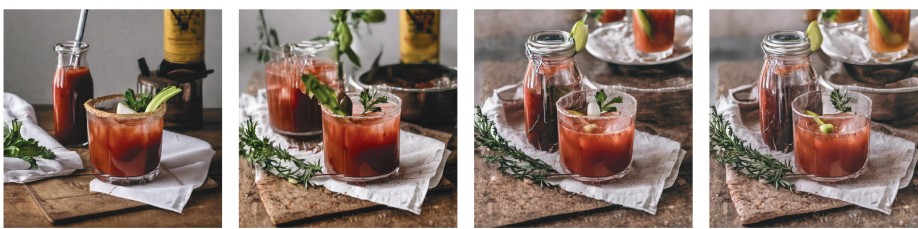

*a bloody mary cocktail next to a napkin*

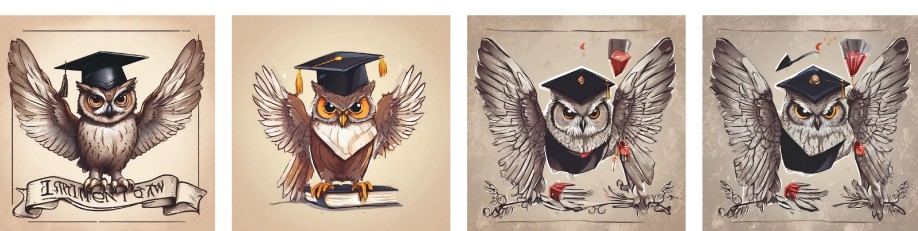

*a cute illustration of a horned owl with a graduation cap and diploma*

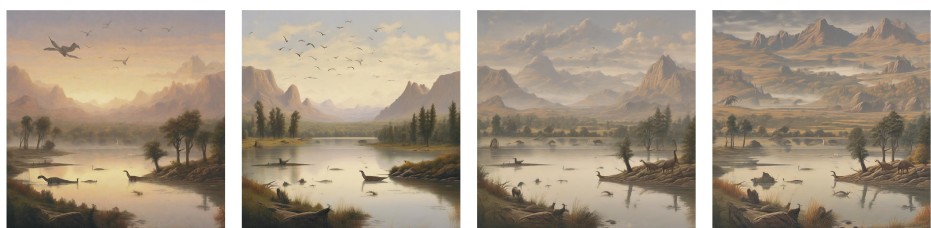

*a peaceful lakeside landscape with migrating herd of sauropods*

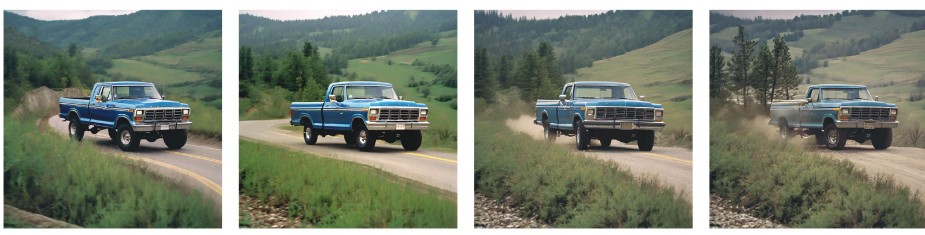

*A blue Ford F-150 coming around a curve in a mountain road*

Figure 17: Generation examples using SDXL, the proposed distilled method using 4 steps and dynamic CFG.

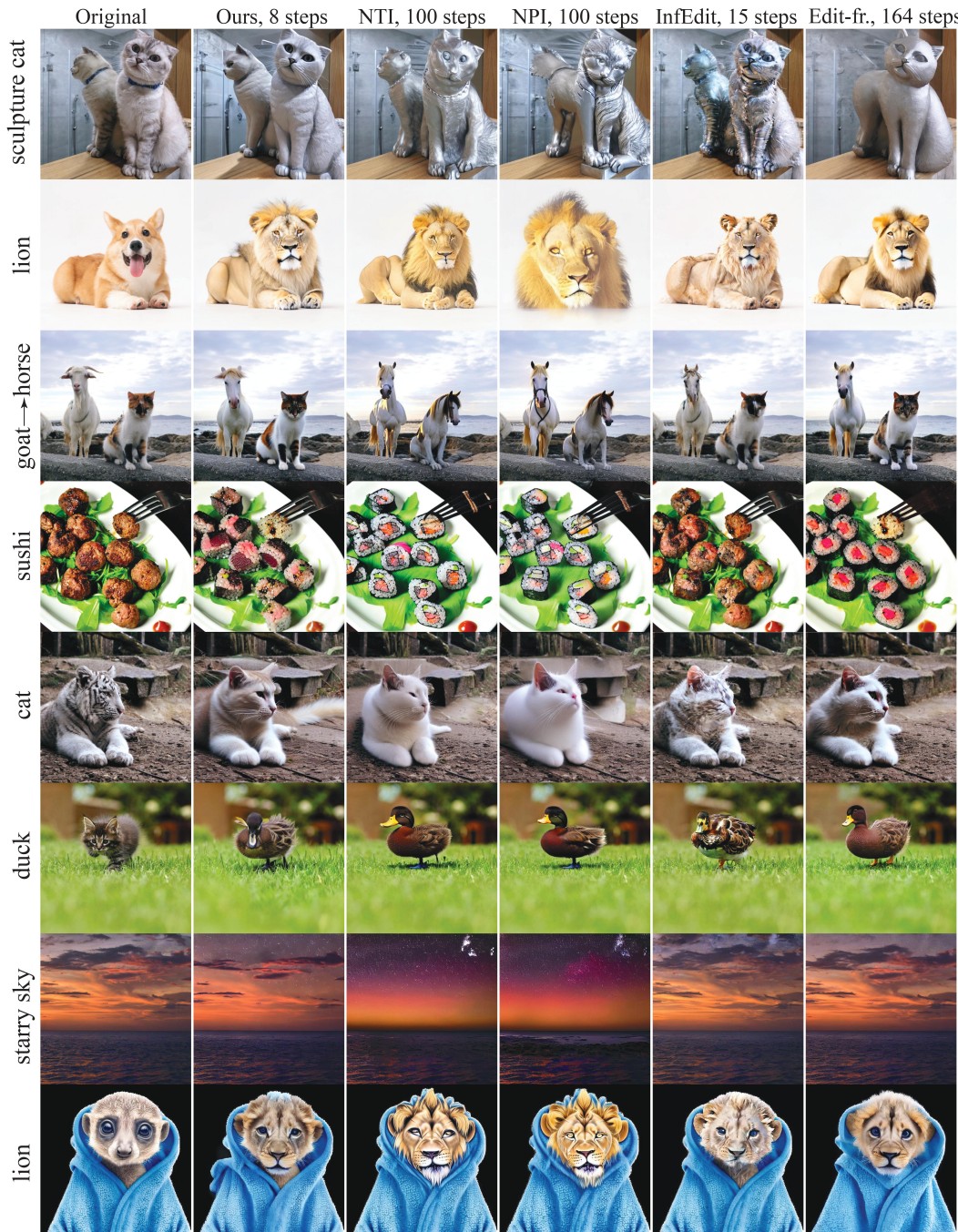

Figure 18: Additional editing examples produced by the SD1.5-based models.

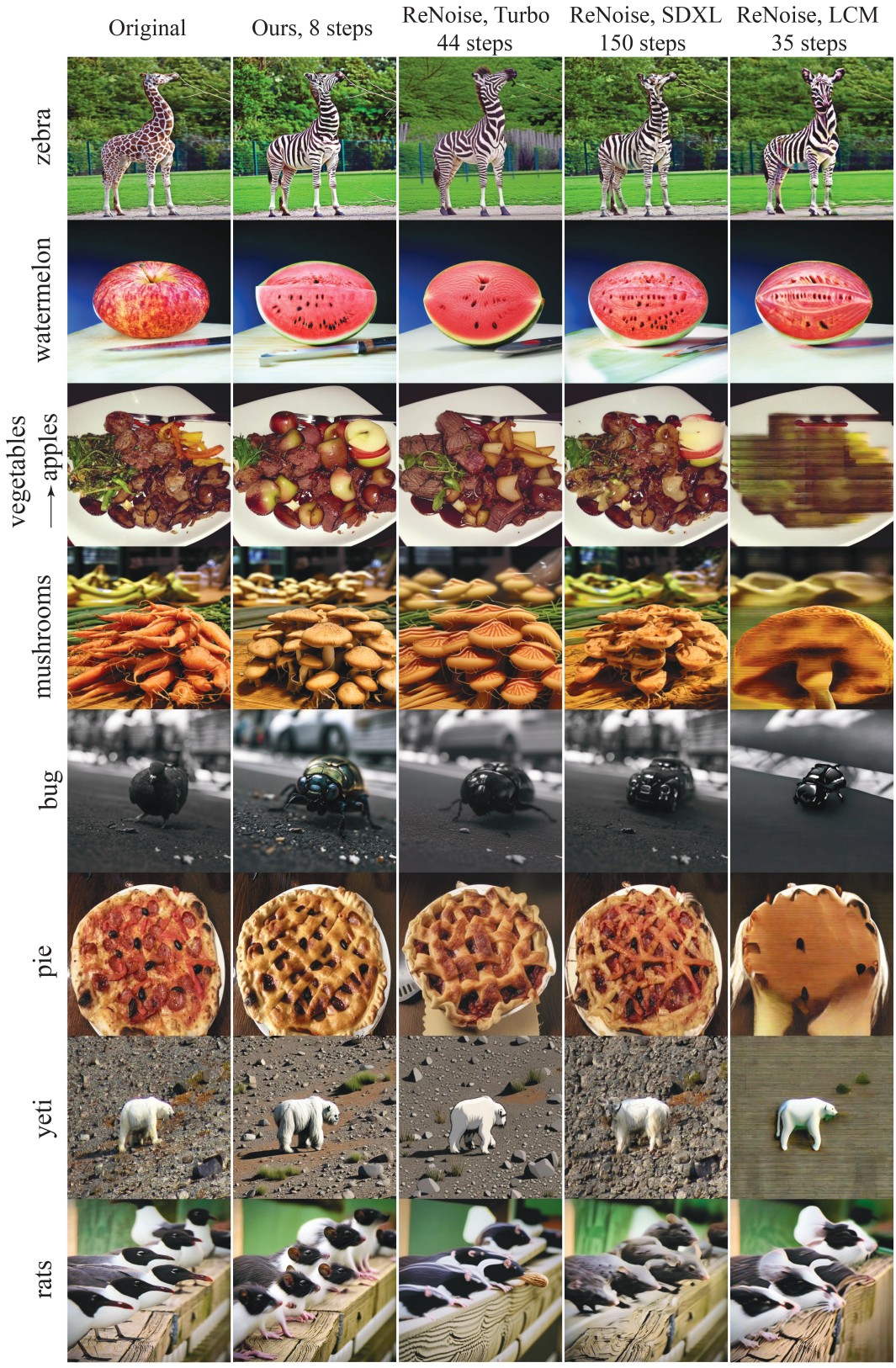

Figure 19: Additional editing examples using the SDXL-based models.

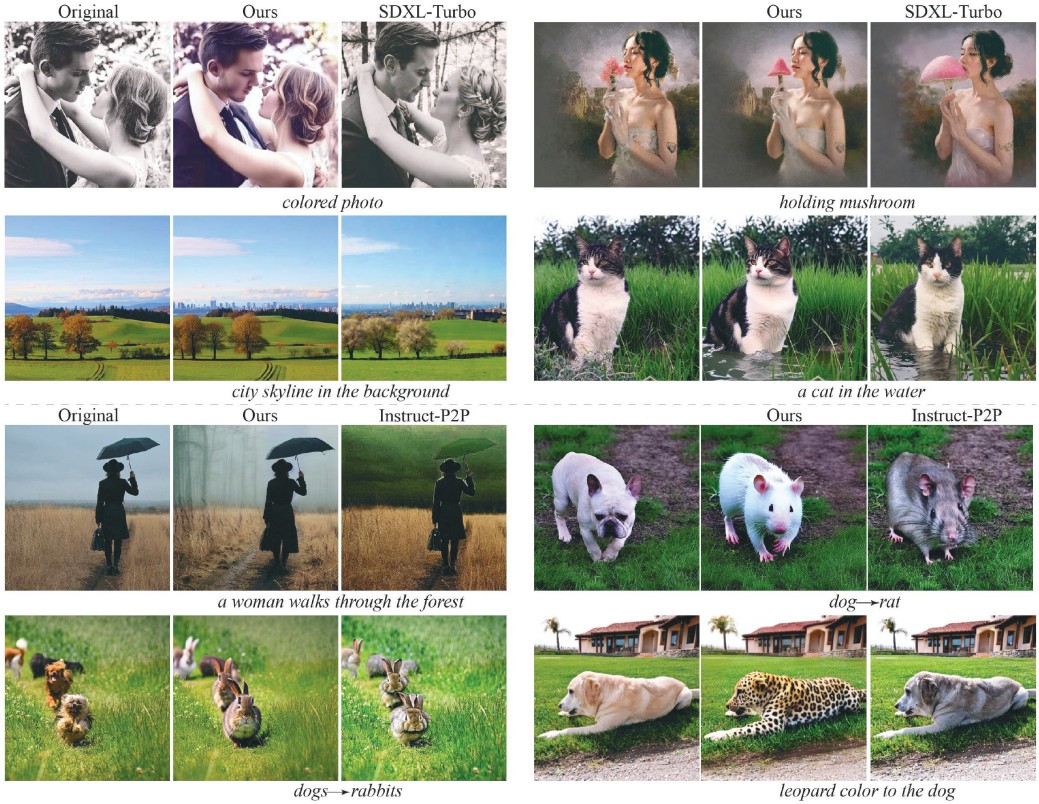

Figure 20: Image editing results produced by our method and SDXL-turbo using SDEdit (upper) and Instruct-P2P (bottom).

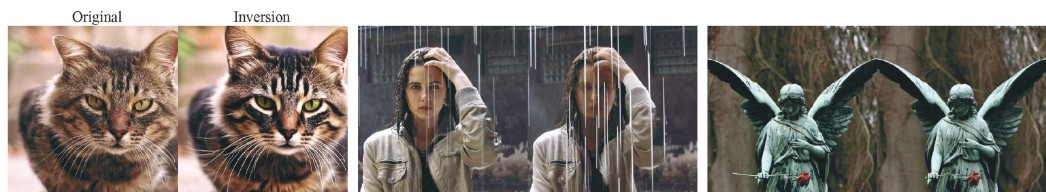

Figure 21: Failure cases.

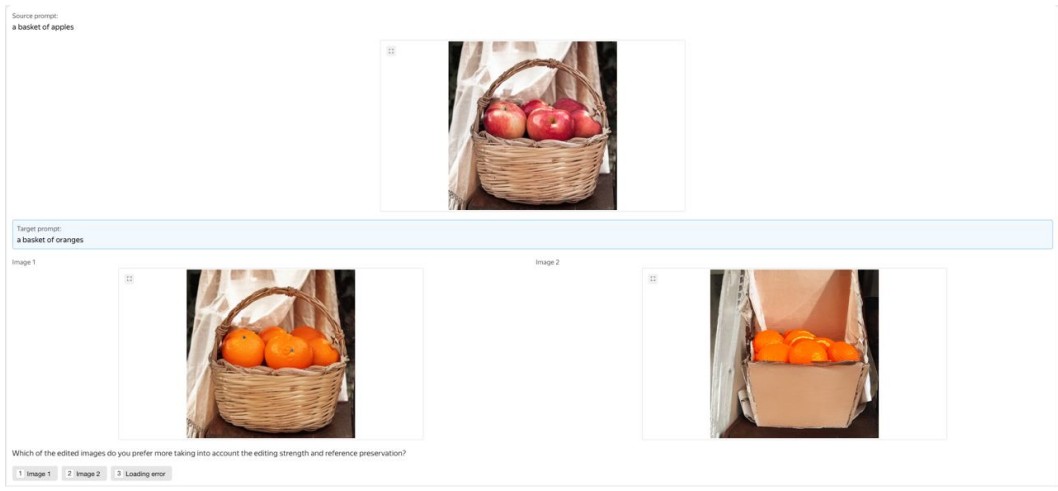

Figure 22: The human evaluation interface for the text-guided image editing problem.

