# OpenReview forum: "Invertible Consistency Distillation for Text-Guided Image Editing in Around 7 Steps"
_NeurIPS.cc/2024/Conference — NeurIPS 2024 poster_

### Official Review · Reviewer_hEvy · 2024-07-11

**Soundness:** 4
**Presentation:** 4
**Contribution:** 4
**Rating:** 6
**Confidence:** 5

**Summary:**

The paper proposes distilation for both of forward and reverse path of ODE sampling of diffusion models in order to enable faster editing.

**Strengths:**

The overall framework is quite interesting, and applying the consistency distillation for forward and reverse process is novel.
Also the method of dynamic CFG is straightforward and seems to have meaningful inversion performance improvement even with CFG.

The proposed method can be combined to various image editing task and expected to show more efficient editing performance.

**Weaknesses:**

1. Although the method shows great performance in image editing with conditions, it feels that the editing method is limited on Prompt2prompt. Is it still possible to apply the model on non-rigid editing such as MasaCTRL?

2. It seems that the most important contributions of this model is faster inversion & sampling. Please explicitly compare the inversion&sampling time with other baseline methods and put the table in main paper.

3. In the experiment part, please compare the model with other training (or fine-tuning) based methods such as instruct-pix2pix or ControlNet. The current experiment only focuses on inversion methods, therefore further experiment would be more helpful for the manuscript.

4. Does the performance degrade if the distillation step decreases? Please give more discussion and experimental results of the case of decreased steps such as single step of 4 steps.

5. minor comment, The Figure 1 (First figure) seems very redundant for me. I recommend a new figure which contains the brief summary of methodology such as combined version of Figure 3 and Figure 1. The figure in the current version make the paper seems unprofessional. As I like the overall idea, it would greatly improve the readability of the manuscript with changing the first figure.

**Questions:**

Already written in the weakness part.

**Limitations:**

Yes

---

> ### Author Rebuttal · Authors · 2024-08-07
>
> Thanks for your careful reading and valuable feedback! We respond to your questions below.
>
> 1. *it feels that the editing method is limited on Prompt2Prompt (P2P). Is it still possible to apply the model on non-rigid editing such as MasaCTRL?*
>
> Our model is not limited to P2P. Alternative editing methods, such as MasaCTRL, are also applicable in the same manner as to original diffusion models. The attached file (Figure 3) provides a few examples of our approach combined with MasaCTRL for non-rigid editing. The results show performance improvement due to more advanced editing methods.  We will add these results in the revision.
> Please note that our work introduces a fast inversion method that is quite orthogonal to editing approaches. Therefore, we evaluate P2P as one of the most well-known approaches to compare various inversion methods fairly.
>
> 2. *Compare the inversion&sampling time with other baselines*
>
> Thanks for the valuable suggestion! We present the time required to invert a single image in the table below. We will add these results to the revision.
>
>
> | Method   | Ours 8 steps, SD1.5     | NTI, SD1.5     |      NPI, SD1.5   | Ours 8 steps, SDXL | ReNoise, LCM-XL
> | -------- | -------- | -------- | -------- | -------- |  -------- |
> | Time, secs     | 0.959+-.005     |116.4+-0.1  | 9.95+-.03      | 1.56+-.07 | 6.75+-.52
>
>
> 3. *Compare the model with other training based methods*
>
> Below, we compare our approach with Intruct-Pix2Pix and observe that it outperforms Intruct-Pix2Pix in terms of both content preservation and editing strength while being training-free. We also present a few visual examples in the attached file (Figure 2, Bottom). We will add this comparison to the revision.
>
> | Method | ImageReward $\uparrow$ | DinoV2 $\uparrow$ | CLIP score, I $\uparrow$ |
> | -------- | -------- | -------- |-------- |
> | Ours, 8 steps    | 0.064     | 0.726     | 0.872 |
> | Instruct-P2P, 100 steps     | -0.227     | 0.708     | 0.850|
>
>
> 4. *Does the performance degrade if the distillation step decreases?*
>
> The scaling performance of our approach is similar to that of consistency distillation in text-to-image generation. Based on our experiments, the optimal number of steps is 6-8 (3-4 encoding + 3-4 decoding steps) and performance noticeably degrades for 2-4 steps (please see the tables below). We believe that future work on consistency models will greatly contribute to improving performance in fewer steps, which can then be transferred to our method.
>
> Inversion:
> | Metrics | Ours, 4+4  | Ours, 2+2 | Ours, 1+1
> | --------  | -------- | -------- | -------- |
> | PSNR $\uparrow$     | 22.81     | 21.91 | 20.85
> | LPIPS $\downarrow$    | 0.179          | 0.235 |  0.244
> | DinoV2 $\uparrow$    | 0.859             | 0.820 | 0.766
>
>
> Editing:
> | Method | ImageReward $\uparrow$ | DinoV2 $\uparrow$ | CLIP score, I $\uparrow$ |
> | -------- | -------- | -------- |-------- |
> | Ours, 8 steps    | 0.064     | 0.726     | 0.872 |
> | Ours, 4 steps     |     -0.035 |   0.553  | 0.815|
> | Ours, 2 steps     | -0.428     | 0.498     | 0.761|
>
>
> 5. *Figure 1 seems very redundant for me*
>
> Thanks for the suggestion! We will update Figure 1 in the revision.

---

> > ### Comment · Reviewer_hEvy · 2024-08-12
> >
> > Thank you for your detailed rebuttal. Most of my concerns have been addressed.

---

### Official Review · Reviewer_Qhsn · 2024-07-11

**Soundness:** 3
**Presentation:** 3
**Contribution:** 2
**Rating:** 5
**Confidence:** 4

**Summary:**

This paper extend the idea of consistency distillation to inversion for image editing. By training a separate consistency model where the consistency is enforced at noise space rather than latent space. Additional cycle consistency loss is employed for more accurate inversion.

**Strengths:**

1. The paper tackles an important problem: efficiency in image inversion. While there are lots of method improving the generation speed of diffusion models, the reverse direction of inversion is less tackled. This paper is an important contribution to this field.

2. The shows promising results in fast image inversion and editing.

**Weaknesses:**

1. The description of the method is unclear. For example, section 3.1 and 3.2 are the main idea of the method, but it ignores an important component: the data. When training consistency distillation, we use perturbed real data, but since here we need to map to the noise, we need to have a coupling of the image and noise. How to obtain this coupling? Do we need to run inversion with the teacher model on a large dataset to obtain the training data? This is very important component of the method but I didn't find an explanation throughout the paper.

2. The method basically needs two models, by employing CD in forward and backward directions, for fast inversion and generation. Although it is practically useful, I feel like it is a bit ad-hoc. The reason is that, if we enable two models, one for fast generation and the other for fast inversion, then any diffusion acceleration methods can be employed separately for two models. It would be more interesting to study this question: given a distilled diffusion model that can be sampled in few steps, can we use this model to do fast inversion?

3. Related to above point. Two models introduce significantly more parameter overhead, and given that inversion based image editing is only only handle a small subset of image editing tasks, it becomes hard to justify whether it worths to double the parameter.

**Questions:**

N/A

---

> ### Author Rebuttal · Authors · 2024-08-07
>
> Thanks for your careful reading and valuable feedback! We respond to your questions below.
>
> 1. *The description of the method is unclear... we need to have a coupling of the image and noise.*
>
> An interesting aspect is that neither additional data nor teacher inversion is required. Compared to the original CD, the only modification needed is the boundary condition. This is possible because the forward CD operates on the same ODE trajectories as the original CD (please see Figure 3 in the paper). The primary difference lies in the direction of the consistency function. Thus, image-noise couplings are "implicitly" created during training using one step of the ODE solver, much like in the original CD. We will provide more details in the revision to ensure clarity.
>
> 2. *Given a distilled diffusion model, can we use this model to do fast inversion?*
>
> It is an interesting question for future research. Since existing distilled models do not support reversibility like diffusion models, we believe that fast inversion of the already distilled models necessiates an extra model to learn image-noise connections. Our intuition is supported by extensive literature on GAN inversion, where an encoder is trained to perform inversion.
>
> 3. *The method basically needs two models...I feel like it is a bit ad-hoc...Two models introduce significantly more parameter overhead....*
>
> We believe that the distilled models need to be redesigned to handle bidirectional sampling (data to noise and noise to data): it can be either an extra input argument, LoRA adapters, or an additional model. As described in Appendix A, our method employs LoRA adapters as a highly convenient and compact way to enable invertibility. Specifically, we use a single diffusion model and simply activate the corresponding LoRA weights depending on which model is used (forward or reverse). Note that LoRA adapters take up a small portion of the total number of parameters (less than 10%). Thus, the parameter overhead is rather negligible compared to storing an extra model.
>
> 4.  *inversion-based image editing only handles a small subset of image editing tasks*
>
> Besides image editing, fast inversion methods can be useful for other problems, such as anomaly detection [1], text-to-3D generation [2,3], image restoration [4].
>
> [1] Hend et al. Out-of-Distribution Detection with a Single Unconditional Diffusion Model
>
> [2] Lukoianov et al. Score Distillation via Reparametrized DDIM
>
> [3] Liang et al. LucidDreamer: Towards High-Fidelity Text-to-3D Generation via Interval Score Matching
>
> [4] Chihaoui et al. Blind Image Restoration via Fast Diffusion Inversion

---

> > ### Author Response · Authors · 2024-08-12
> >
> > Dear Reviewer Qhsn,
> >
> > Given the limited time for discussion, we would appreciate it if you could let us know whether we have fully addressed your concerns and whether there are any other questions we might need to address.

---

> ### Comment · Reviewer_Qhsn · 2024-08-12
> **After rebuttal**
>
> Thanks authors for the response.
>
> Now I am clear about the data you are using. But I am still not convinced that if it worths to have two models just for fast inversion. I will increase my score to 5 to reflect your clarification.

---

### Official Review · Reviewer_QtT7 · 2024-07-23

**Soundness:** 3
**Presentation:** 3
**Contribution:** 3
**Rating:** 6
**Confidence:** 4

**Summary:**

This work introduces invertible Consistency Distillation (iCD), which enhances text-to-image diffusion models by enabling effective encoding of real images into latent space. iCD achieves both high-quality image synthesis and accurate image inversion in just 3-4 inference steps.

**Strengths:**

1. The adaptation of Consistency Distillation to tackle image editing tasks represents a valuable and promising research direction.
2. The design of the forward and backward consistency distillation mechanisms is interesting.
3. The experimental results are impressive, demonstrating that the proposed method achieves similar or even superior editing effects in fewer inference steps compared to existing models.

**Weaknesses:**

1. Discussion of Failure Cases: It would be beneficial to include a detailed discussion of failure cases. Understanding the scenarios where the method does not perform optimally can provide valuable insights and guide future research directions.

2. More SDXL Demos Needed: The main paper (Figure 6) showcases demos using SD1.5, which, however, still suffers from generation issues like details of hands and faces. It is better to include more examples using SDXL (which offers improved performance) in the main paper. This would strengthen the evaluation and illustrate the capabilities of the proposed method more effectively.

3. Comparison with SDXL Turbo (Adversarial Distillation): SDXL turbo is known for its excellent performance in one-step generation and can be used for editing with tools like SDedit. A comparative analysis of the editing effectiveness and efficiency between the proposed method and adversarial distillation methods would provide a clearer understanding of the advantages and potential trade-offs, offering a more comprehensive evaluation of the proposed approach.

**Questions:**

Please refer to the weaknesses.

**Limitations:**

This work has discussed limitations in Appendix.

---

> ### Author Rebuttal · Authors · 2024-08-07
>
> Thanks for your careful reading and valuable feedback! We respond to your questions below.
>
> 1. *Discussion of Failure Cases*.
>
> We present some inversion failure cases in the attached file (Figure 1). Our method sometimes oversaturates images for high guidance scales and struggles to reconstruct complex details like human faces and hands. We will add the discussion and more examples in the revision.
>
> 2. *More SDXL Demos Needed*.
>
> The paper has SDXL illustrations in figures 8 and 16. To strengthen the evaluation, we will provide more SDXL examples in the revision.
>
> 3. *Comparison with SDXL Turbo using SDEdit*.
>
> Thanks for the important suggestion! Please see the comparisons in the table below and the qualitative results in the attached file (Figure 2, Top). SDXL Turbo with SDEdit significantly hurts reference image preservation due to stochasticity, as confirmed by the DinoV2, image CLIP score and qualitative comparisons. This highlights the importance of accurate image inversion for editing. Moreover, our approach outperforms SDXL Turbo in terms of editing strength (ImageReward).
>
>
> | Method | ImageReward $\uparrow$ | DinoV2 $\uparrow$ | CLIP score, I $\uparrow$ |
> | -------- | -------- | -------- |-------- |
> | Ours     | 0.473     | 0.726     | 0.873 |
> | SDXL-turbo     | 0.364     | 0.637     | 0.835|

---

> > ### Author Response · Authors · 2024-08-12
> >
> > Dear Reviewer QtT7,
> >
> > Given the limited time for discussion, we would appreciate it if you could let us know whether we have fully addressed your concerns and whether there are any other questions we might need to address.

---

> > ### Comment · Reviewer_QtT7 · 2024-08-12
> >
> > Thank you for the rebuttal. I have no further concerns. Adding these detailed discussions and comparisons will undoubtedly enhance the quality of the paper.

---

### Official Review · Reviewer_H4g5 · 2024-07-23

**Soundness:** 3
**Presentation:** 3
**Contribution:** 2
**Rating:** 7
**Confidence:** 3

**Summary:**

This paper targets a novel problem of enabling image inversion and editing for models distilled with consistency distillation.
The authors identify the challenges of applying consistency distillation, which is designed for the denoising process, to the diffusion process and propose multi-boundary consistency distillation that achieves inversion with as few as 3-4 steps. A regularization term is proposed to guarantee the consistency between the forward and backward models. The authors further study dynamic CFG for performance improvements. The effectiveness is validated on SD 1.5 and SDXL models.

**Strengths:**

- This paper targets a very emerging and important problem of how to adapt the abilities DM models have to the accelerated version such as consistency-distilled models.

- The paper is generally well-organized, and the presentation is easy to follow.

- The empirical results are pretty strong given the very small number of steps used by the proposed method.

**Weaknesses:**

- While the overall problem is novel, and the solution demonstrates strong performance, almost every individual technique presented in this paper comes from another earlier work and makes the overall contribution less impactful.

- The proposed framework is highly customized toward CD, and might not be able to generalize other DM model acceleration methods such as SD-turbo, DMD, and UFOGen.

**Questions:**

The presented evaluations seem to focus primarily on 7 steps. While the performance is impressive compared to other methods, the higher performance reported by fDDIM50 makes me wonder if the proposed method can match the performance with more steps and what the marginal performance would be if we scale up the number of steps.

Minor suggestion, doesn't affect rating:
The overall quality of this paper can be greatly improved by replacing Figure 1. The current one seems too simple and needs to be more informative. Figure 6 can even be a better alternative for Figure 1.

**Limitations:**

The authors have properly discussed the limitations in Appendix Section E.

---

> ### Author Rebuttal · Authors · 2024-08-07
>
> Thanks for your careful reading and valuable feedback! We respond to your questions below.
>
> 1. *almost every technique comes from another earlier work...contribution less impactful.*
>
> Our techniques present a natural continuation of previous works without compromising novelty, in our opinion. Invertible Сonsistency Distillation and its corresponding training pipeline have not been formulated before, and dynamic guidance was not explored from the inversion perspective. We believe our approach makes an important contribution to the field, as it can be useful for future works on inversion-based problems.
>
> 2. *The proposed framework is highly customized toward CD, not be able to generalize SD-turbo, DMD, and UFOGen.*
>
> Unlike GAN- or DMD-based distillation methods, CD exploits the ODE perspective of diffusion models. In our ablation study in Table 2E, we show that using ODE perspective via CD objective largely enhances our inversion method. The inversion of other distillation methods remains an open and interesting question.
>
> 3. *Can the proposed method match the performance with more steps and what the marginal performance would be if we scale up the number of steps?*
>
> In the table below, we present the results of the improved version of our model after more careful tuning. 8 steps of our inversion closely approaches the performance of the teacher inversion. Therefore, further scaling does not seem necessary. We will provide more details in the revision.
>
> | Metrics | Ours, 4+4  | DDIM, 50+50 |
> | -------- | -------- | -------- |
> | PSNR $\uparrow$     | 22.81  | 23.07     |
> | LPIPS $\downarrow$    | 0.179         | 0.167   |
> | DinoV2 $\uparrow$    | 0.859         | 0.851     |
>
> 4. *The overall quality of this paper can be greatly improved by replacing Figure 1*.
>
> Thanks for the suggestion! We will update Figure 1 in the revision.

---

> > ### Author Response · Authors · 2024-08-12
> >
> > Dear Reviewer H4g5,
> >
> > Given the limited time for discussion, we would appreciate it if you could let us know whether we have fully addressed your concerns and whether there are any other questions we might need to address.

---

### Author Rebuttal · Authors · 2024-08-07

We would like to thank the reviewers for their constructive feedback, which will help us significantly improve our work. In our individual responses, we address the raised questions and concerns and we attach a PDF file with supporting qualitative results.

---

### Decision · Program_Chairs · 2024-09-25

**Decision:**

Accept (poster)

**Comment:**

This paper extends distilled text-to-image diffusion models to fast editing by introducing invertible consistency distillation, which allows efficient image inversion and synthesis. All reviewers give positive scores after rebuttal. The problem is interesting, the method is novel, and the performance is significant. AC agrees this is a nice contribution on diffusion models and image editing, and thus recommends acceptance.